# Ionic selectivity and thermal adaptations within the voltage-gated sodium channel family of alkaliphilic *Bacillus*

Paul G DeCaen[1], Yuka Takahashi[2,3], Terry A Krulwich[4], Masahiro Ito[2,3], David E Clapham[1,5]*

[1]Department of Cardiology, Howard Hughes Medical Institute, Boston Children's Hospital, Boston, United States; [2]Graduate School of Life Sciences, Toyo University, Gunma, Japan; [3]Japan and Bionano Electronics Research Center, Toyo University, Saitama, Japan; [4]Department of Pharmacology and Systems Therapeutics, Icahn School of Medicine at Mount Sinai, New York, United States; [5]Department of Neurobiology, Harvard Medical School, Boston, United States

**Abstract** Entry and extrusion of cations are essential processes in living cells. In alkaliphilic prokaryotes, high external pH activates voltage-gated sodium channels ($Na_v$), which allows $Na^+$ to enter and be used as substrate for cation/proton antiporters responsible for cytoplasmic pH homeostasis. Here, we describe a new member of the prokaryotic voltage-gated $Na^+$ channel family (NsvBa; Non-selective voltage-gated, *Bacillus alcalophilus*) that is nonselective among $Na^+$, $Ca^{2+}$ and $K^+$ ions. Mutations in NsvBa can convert the nonselective filter into one that discriminates for $Na^+$ or divalent cations. Gain-of-function experiments demonstrate the portability of ion selectivity with filter mutations to other *Bacillus* $Na_v$ channels. Increasing pH and temperature shifts their activation threshold towards their native resting membrane potential. Furthermore, we find drugs that target *Bacillus* $Na_v$ channels also block the growth of the bacteria. This work identifies some of the adaptations to achieve ion discrimination and gating in *Bacillus* $Na_v$ channels.

**\*For correspondence:**
dclapham@enders.tch.harvard.edu

**Competing interests:** The authors declare that no competing interests exist.

## Introduction

Ion selectivity is a defining feature of ion channels. While the structural and biophysical determinants of $K^+$ selectivity are well described (*Doyle et al., 1998*; *Jiang et al., 2003*), those of $Na^+$ and $Ca^{2+}$ are unresolved. To fill this knowledge gap, the voltage-gated sodium channels ($Na_v$ channels) from bacteria have been extensively studied by structural biologists. To date, two full-length (NavAb and NavRh) (*Payandeh et al., 2011, 2012*; *Zhang et al., 2012*) and three pore-only (NavMs, NavAe and NavCt) (*McCusker et al., 2012*; *Tsai et al., 2013*; *Shaya et al., 2014*) prokaryotic $Na^+$ channel crystal structures have been solved. The full-length structures demonstrate that four identical subunits, each containing 6-transmembrane segments, assemble together to form a functional channel. The first four transmembrane segments form the voltage sensor (S1–S4) while the fifth and sixth transmembrane segments (S5, S6) form the pore domain. The selectivity filter, which forms critical interactions with the permeating hydrated ions, defines the pore and is scaffolded by two pore helices (P1 and P2) from each subunit. The dipole created by the helices, and an acidic residue from each subunit, create an electronegative region that attracts cations. However, the molecular arrangement that enables $Na^+$-selectivity in prokaryotes might differ from mammalian voltage-gated $Na^+$ channels. Thus, it is not clear if structural features that determine ion selectivity in homotetrameric prokaryotic $Na_v$ reflect the functionally heterotetrameric eukaryotic sodium channels.

The first prokaryotic voltage-gated sodium channel, cloned from *Bacillus halodurans* C-125, was called NaChBac (Na+ Channel of Bacteria) (*Ren et al., 2001*). Since then, $Na_v$ channels from at least

**eLife digest** Life essentially runs on electricity: electrical signals cause nerve cells to fire, heart muscles to contract and allow organisms to sense the world around them. These signals are triggered by the movement of positively-charged ions—such as sodium, potassium and calcium—moving into a cell through special ion channels in the cell membrane, which can open and close in response to changes in the voltage across the cell membrane.

With few exceptions, voltage sensitive ion channels usually only let one type of ion pass into the cell. But how do ion channels discriminate amongst ions and how did they acquire this ability during evolution? To address these questions, researchers have studied a family of sodium channels from bacteria for the past decade. Here DeCaen et al. describe a new member from this ion channel family from a bacterium called *Bacillus alcalophilus*. This ion channel does not discriminate between positively-charged ions and *B. alcalophilus* needs this ion channel for it to dwell in environments that have high levels of potassium or sodium. DeCaen et al. demonstrate that these ion channels can be made selective for sodium or calcium with as little as two small changes in the gene that encodes the ion channel. Furthermore, making similar genetic mutations in related ion channel genes from other *Bacillus* species has the same effect. DeCaen et al. suggest that *Bacillus* ion channel genes are easily adapted to function in a variety of environmental conditions with different levels of positively-charged ions. Thus it is easier for *Bacillus* channels to evolve to be selective for different ions.

*Bacillus* bacteria divide rapidly in warm to hot temperatures and under alkaline pH. DeCaen et al. demonstrate that both of these conditions make *Bacillus* ion channels easier to open in response to voltage. In addition, DeCaen et al. demonstrate that *Bacillus* ion channels can be targeted by drugs that impair the ability of the bacteria to grow. These findings—together with other work that revealed where drug molecules bind to ion channels—could potentially guide efforts to develop treatments for illnesses caused by other *Bacillus* strains, which include anthrax and some forms of food poisoning.

9 bacterial species have been functionally characterized. All full-length channels exhibit $Na^+$-selectivity (*Ren et al., 2001*; *Ito et al., 2004*; *Koishi et al., 2004*; *Irie et al., 2010*; *Ulmschneider et al., 2013*). Most bacteria require the ion-motive forces provided by $H^+$ or $Na^+$ ions that move through the MotAB or MotPS stator to power the rotary motor proteins to drive the motion of the flagella, (*Figure 1A*) (*Ito et al., 2004*; *Fujinami et al., 2009*). In soda lakes and other extremely $H^+$-poor (pH 9–12; 1 nM—1 pM $[H^+]$) and $Na^+$-rich environments (up to 500 mM), alkaliphilic prokaryotes couple motility and ion trans-porters to $Na^+$ rather than $H^+$. Growth and motility of the alkaliphilic bacterium *Bacillus alcalophilus* AV (*Vedder, 1934*) are better supported in $K^+$ rather than $Na^+$ conditions (*Terahara et al., 2012*). In alka-liphilic *Bacillus*, either ion can drive the flagellar motor, and a single mutation in the MotS protein was identified that converted the naturally nonselective MotPS ($Na^+$ or $K^+$) stator into a $Na^+$-selective stator. Presumably, bacterial Nav channels provide a source of $Na^+$ ions that drives the stators and maintains ion homeostasis. However, it is unclear what condition or stimulus would open these channels at the very hyperpolarized resting membrane potentials ($\Psi_{rest} \approx -180$ mV) found in bacteria.

Here, we determine the permeation and gating properties of a voltage-gated ion channel (NsvBa) from *Bacillus alcalophilus* AV. While related to members of the $Na_v$ superfamily (*Figure 1B*), NsvBa is nonselective among cations. Using mutagenesis, we demonstrate that the nonselective filter from NsvBa can be converted into one that is more selective for $Na^+$ or $Ca^{2+}$, and that these features can be conferred onto the NaChBac filter. In addition to selectivity, we characterize the voltage-, pH- and temperature-dependence of *Bacillus* $Na_v$ channels. We find that a combination of higher temperature and pH are required to reduce the activation threshold of channel opening in *Bacillus* $Na_v$ channels, which is unique to this family of sodium channels. We also characterize $Na_v$ current antagonism by drugs that impaired the growth and motility of alkaliphilic *Bacillus* species at corresponding concentrations. These findings shed light on the biophysical requirements for ion channel selectivity, pharmacology, biochemical adap-tations among *Bacillus* species, and the evolution of voltage-gated $Na^+$ channels.

## Results

Among the first alkaliphilic extremophiles described in the literature, the gram-positive, rod-shaped *Bacillus alcalophilus* AV was initially isolated from human feces (*Vedder, 1934*). We cloned NsvBa and

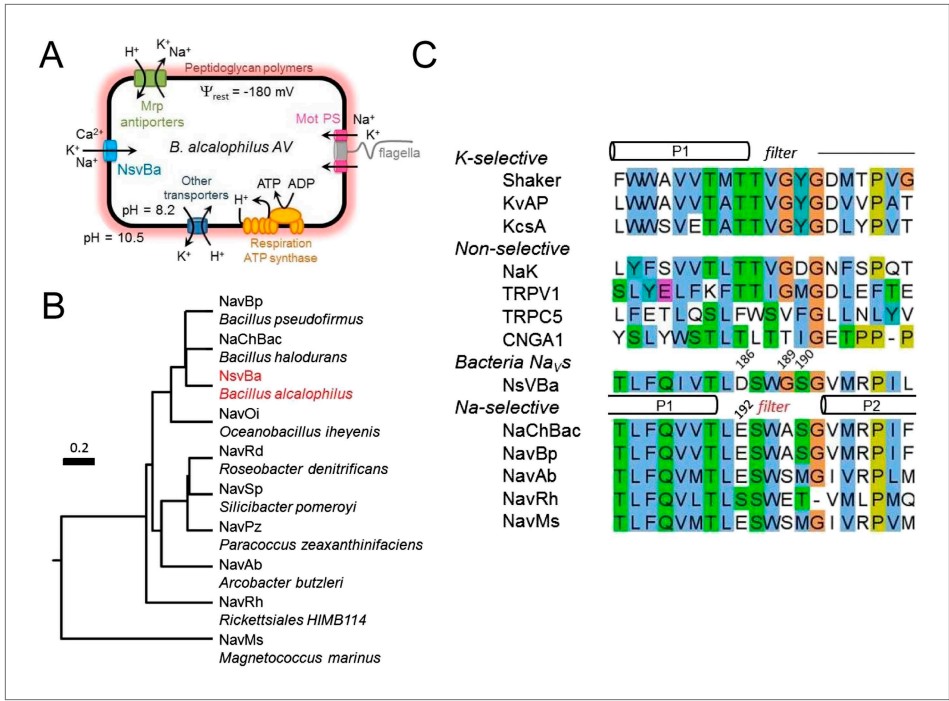

**Figure 1**. The alkaliphilic *Bacillus* cation cycle and the relationship between bacteria Na$_v$ homologs. (**A**) A diagram depicting the membrane proteins involved in the Na$^+$ cycle of *Bacillus alcalophilus*. The cation/proton antiporters, including Mrp antiporters, catalyze net proton accumulation in the cytoplasm in cells that are extruding protons during respiration. Na$^+$ re-entry in support of pH homeostasis is achieved by Na$^+$ solute symporters and through the voltage-gated channel, NsvBa . (**B**) Relatedness within functionally characterized members of the bacterial sodium channel superfamily. A rooted phylogenic tree analysis of bacterial Na$_v$ channels calculated by using the CLUSTALW program (http://clustalw.genome.jp). Branch lengths are proportional to the sequence divergence (scale bar = 0.2 substitutions per amino acid site). (**C**) An alignment of the selectivity filters from various Na$^+$, K$^+$, and nonselective ion channels. The pore region from the indicated ion channel families (*italic*) were aligned using the ClustalW multiple sequence alignment program applying the default color scheme with <60% conservation of character: Hydrophobic (blue); polar (green); glutamine, glutamate, aspartate (magenta). Special amino acids are designated with their own color: glycine (orange); proline (yellow) and tyrosine or histidine (cyan). The barrels indicate the regions spanning the pore helices found in the *Shaker* (K$^+$-selective), NaK and NavRh crystal structures.

The following figure supplement is available for figure 1:

**Figure supplement 1**. A comparison of the voltage-dependence of the bacterial Na$_v$ channels.

generated plasmids for mammalian cell expression to enable measurement by patch clamp methods. Currents from NsvBa-transfected HEK293T cells were robust (≈119 pA/pF in 150 mM extracellular [Na$^+$]) with voltage-dependent activation and inactivation similar to NaChBac (*Figure 1—figure supplement 1A,B*). The time constant of NsvBa Na$^+$ current inactivation ($\tau_{inact}$) measured at 0 mV was 42 ms, ~2 times faster than NaChBac (78 ms) but ~6 times slower than NavMs (7 ms, *Figure 1—figure supplement 1C*). The sequence TLESWxxG is conserved in the selectivity filters of prokaryotic Na$^+$ channels (*Figure 1C*). Although the sequence of NsvBa is homologous to other prokaryotic Na$_v$ channels, the selectivity filter sequence (TLDSWGSG) deviates at the high field-strength site (Site$_{HFS}$ most extracellular site; see Discussion) with an aspartate residue replacing glutamate (D186) and a flexible glycine at an extracellular position (G189). When aligned, G189 and G191 form a 'GxG motif' commonly found in K$^+$-selective (Kv) and nonselective ion channels (*e.g.*, TRP and CNG channels, *Figure 1C*). To determine the selectivity of the NsvBa channel, we patch clamped transiently-transfected HEK cells and measured voltage-dependent currents in the presence of monovalent alkali ions (Li$^+$, Na$^+$, K$^+$, Rb$^+$, Cs$^+$) or divalent alkaline earth metals (Mg$^{2+}$, Ca$^{2+}$, Sr$^{2+}$, Ba$^{2+}$) (*Figure 2A,B*). Compared to the relatively Na$^+$-selective NaChBac channel, NsvBa was nonselective and all cations but Cs$^+$ and Mg$^{2+}$ permeated the pore (*Figure 2C,D*). NsvBa Na$^+$ and K$^+$ single channel conductances were equivalent (30 ± 3 pS

and 36 ± 3 pS, respectively, *Figure 2—figure supplement 1*), indicating that the channel does not distinguish between these ions.

Given the 61% shared amino acid identity between NaChBac and NsvBa channels, we were able to introduce the NaChBac selectivity filter (TLESWASG) into the NsvBa channel to examine two relevant residues, D186E and G189A. These changes conferred Na$^+$-selectivity onto the channel and abrogated all other monovalent and divalent conductances (*Figure 3A,B*). When tested separately, the single mutation D186E was sufficient for Na$^+$-selectivity (*Figure 3—figure supplement 1A,B*), whereas the single mutation of G189A substantially reduced, but did not entirely abolish, the permeability of K$^+$ and Rb$^+$ (*Figure 3—figure supplement 1C,D*). Furthermore, the G189A mutant channel still conducted Ca$^{2+}$ and Ba$^{2+}$ divalent ions, whereas the D186E mutation did not. These data suggest that a glutamate at Site$_{HFS}$ selects Na$^+$ among other cations while the 'GxG motif' likely provides an extracellular K$^+$ or Rb$^+$ ion coordination site, possibly involving the glycine backbone carbonyls as found in the selectivity filter of the potassium and NaK channels (*Doyle et al., 1998*; *Jiang et al., 2002*; *Alam and Jiang, 2009a*; *Alam and Jiang 2009b*). Next we sought to determine whether the NsvBa filter sequence could be transferred into Na$^+$-selective NaChBac. When the Na$^+$-selectivity filter of NaChBac (TLESWASG) was mutated at two positions (E192D:A195 G) to conform to the NsvBa nonselective filter (TLDSWGSG), ion selectivity was identical to that of NsvBa (*Figure 3C,D*). This finding illustrates the mutual portability of selectivity between the voltage-gated cation channels in *Bacillus* species *halodurans* and *alcalophilus* AV.

The selectivity for Na$^+$ ions in vertebrate Na$_v$ channels is attributed to an asymmetric ring of 4 amino acids (Asp, Glu, Lys, and Ala: DEKA) contributed by each of the pore-lining loops of the 4 domains (*Catterall, 2012*). Voltage gated calcium channels (Ca$_v$s) are thought to achieve Ca$^{2+}$-selectivity by a symmetric ring of 4 glutamate residues (EEEE), each contributed by one domain of the polypeptide (*Hess et al., 1986*). In contrast, prokaryotes achieve Na$^+$-selectivity from an apparent 4-fold symmetry of acidic residues, each from a subunit in the homomer. In previous studies, we demonstrated that the Na$^+$-selective NaChBac (TLESWASG) filter could be converted into one that prefers the divalents Ba$^{2+}$ and Ca$^{2+}$ by introducing acidic residues into three positions in the filter sequence (TLDDWADG) (*Yue et al., 2002*). This filter sequence also was grafted into the NavAb channel (called CavAb), shown to be more Ca$^{2+}$-selective, and the high-resolution structure determined (*Tang et al., 2014*). We tested the LDDWADG mutated filter (S187D:G189A:S190D) in the NsvBa channel and confirmed that it was divalent permeant (Ca$^{2+}$, Sr$^{2+}$, Ba$^{2+}$) but it also had measurable permeability to monovalent cations (including K$^+$ and Rb$^+$), demonstrating that Ca$^{2+}$-preference achieved by this filter (P$_{Ca}$/P$_{Na}$~30, *Figure 4*; *Figure 4—figure supplement 1*) is much lower than mammalian Ca$_v$ channels (P$_{Ca}$/P$_{Na}$ ≥ 1000) (*Tsien et al., 1987*). Rather, this selectivity is more analogous to some members of the TRP channel family, such as TRPV5, TRPV6 (*Owsianik et al., 2006*; *Wu et al., 2010*). A comparison of the filter mutations effects on relative permeability of the NaChBac and NsvBa channels are summarized in *Figure 4* and listed in *Figure 4—figure supplement 2*. We also attempted to convert the NsvBa channel into a K$^+$-selective channel by changing the selectivity filter sequence (TLTSWGSG and TLTSWGYG), but these channels either did not express on the plasma membrane or did not conduct cations under our experimental conditions (data not shown).

Alkaliphilic *Bacillus* are estimated to have very negative resting membrane potentials ($\Psi_{rest}$ ≈ −180 mV), although membrane potentials in bacteria are measured from voltage- sensitive dye studies and variability within populations can be large. Nevertheless, the activation threshold for *Bacillus* sodium channels is ≈ −40 mV, which is extremely depolarized relative to estimates of $\Psi_{rest}$. Since alkaliphilic bacteria live in high pH environments, we tested whether their sodium channel gating shifted as a function of pH. As shown for the Na$^+$-selective channel from *Bacillus pseudofirmus* OF4 (NavBp) (*Ito et al., 2004*), Na$^+$ currents from NaChBac and NsvBa are also modulated by high extracellular pH (*Figure 5*). When extracellular pH was increased from 7.4 to 9.4, the peak current increased twofold to fourfold and the steady state voltage-dependence was negatively shifted by 28–34 mV (*Figure 5—figure supplement 1*). Basic extracellular pH alone is probably insufficient to reduce this substantial energy barrier to activate these channels from $\Psi_{rest}$ = −180 mV (≈−3.2 kcal/mol). Thus additional influences are required to bring $\Psi_{rest}$ and V$_{1/2}$ closer together.

Many alkaliphilic *Bacillus* species growth rates are temperature-dependent (30–60°C). Thus we tested the effects of temperature (20–37°C) at neutral and basic extracellular pH (7.4 and 9.4, respectively) on the Na$^+$ currents conducted by *Bacillus* NsvBa, NaChBac and NavBp channels (*Figure 6A–C*). At neutral pH, we observed that increasing the temperature shifted the voltage dependence of

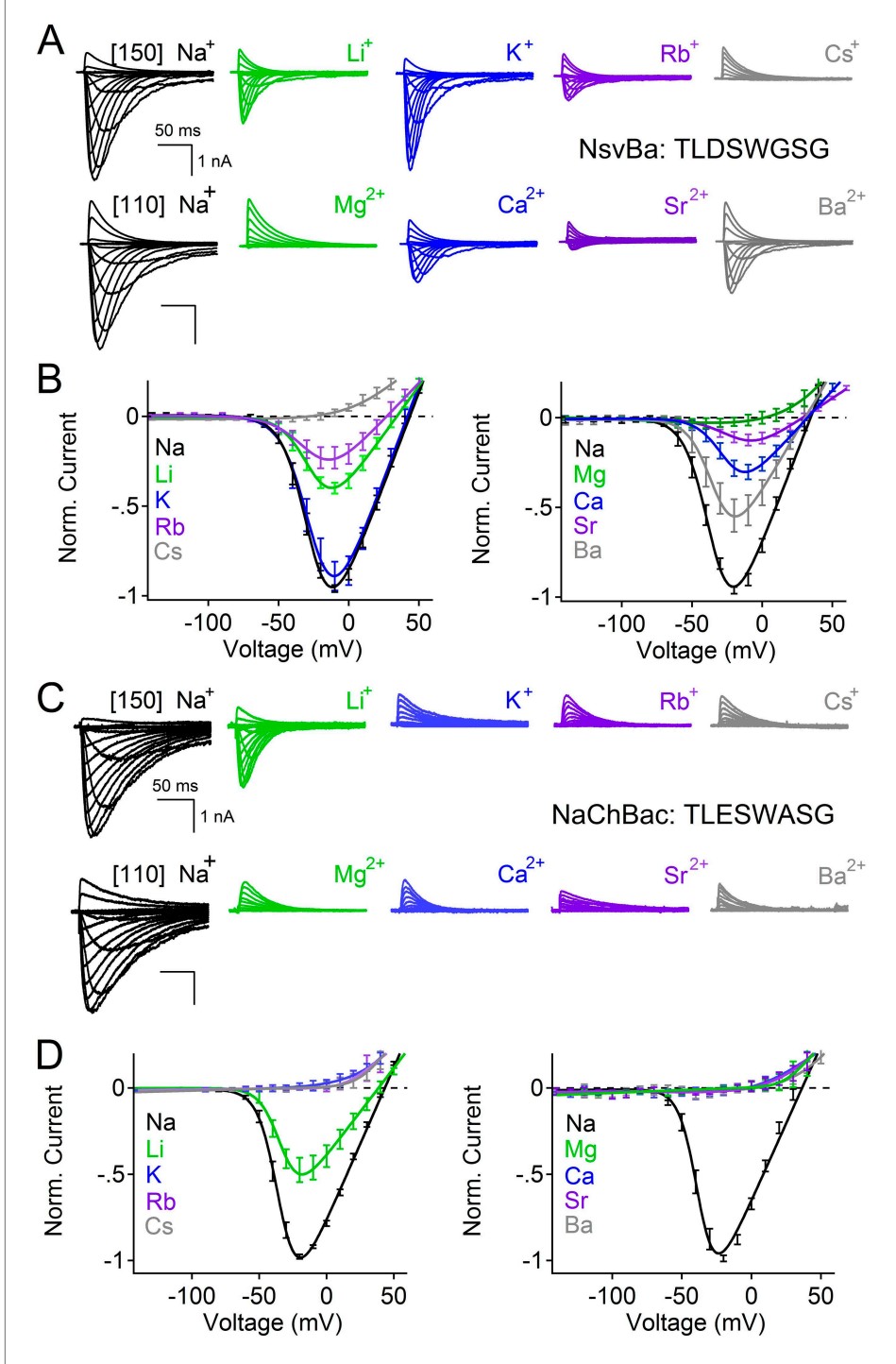

**Figure 2**. Comparison of cation selectivity between the nonselective NsvBa and Na⁺-selective NaChBac channels. (**A** and **C**) Representative current traces from NsvBa channel (**A**) or from the NaChBac channel (**C**) showing the first 0.25 s of 0.5 s activations from −180 mV holding potential: *Top,* 150 mM Na⁺ was substituted with an equal concentration of the indicated monovalent ions; *Bottom,* 110 mM Na⁺ was substituted for equal concentrations of the indicated divalent cations. (**B** and **D**) Resulting current–voltage relationships measured for the conditions tested in (**A**) and (**C**). NsvBa: n = 5–9, NaChBac: n = 6–9; Error = ±SEM.

The following figure supplement is available for figure 2:

**Figure supplement 1**. Na⁺ and K⁺ are highly conductive through the nonselective NsvBa channel.

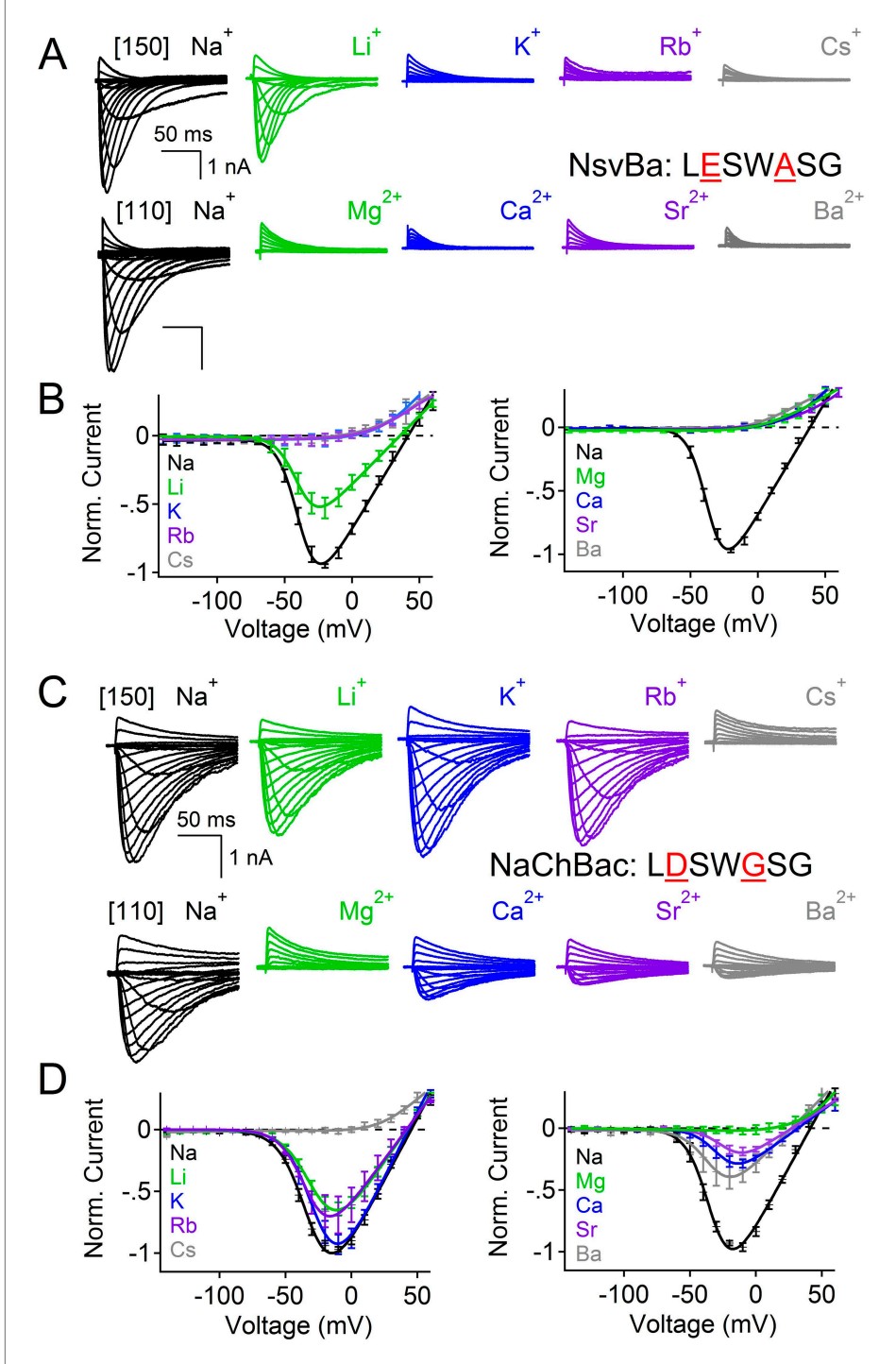

**Figure 3**. Reciprocal substitutions of the NsvBa and NaChBac filters transfers cation selectivity. (**A**, **B**) The selectivity mutant NsvBa channel containing the NaChBac selectivity filter sequence TLESWASG (D186E: G189A) or the (**C**, **D**) mutant NaChBac channel containing the NsvBa selectivity sequence TLDSWGSG (E192D:A195G). (**A** and **C**) Representative current traces from the mutant channels under the same conditions described in *Figure 2*. (**B** and **D**) Resulting current-voltage relationships measured for the mono- and divalent conditions. (n = 4–9 for both channels, Error = ±SEM).

The following figure supplement is available for figure 3:

**Figure supplement 1**. The effects of the single mutations D186E and G189A in the NsvBa selectivity filter.

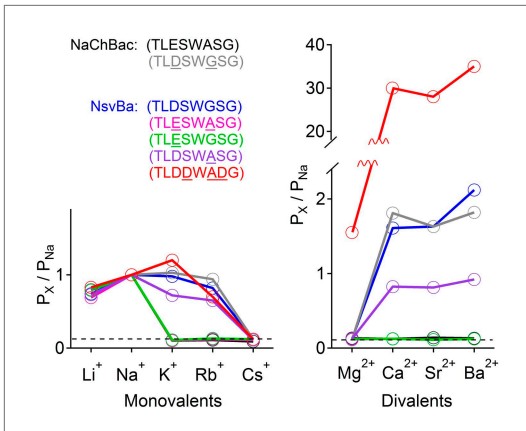

**Figure 4**. Summary of the relative permeability of cations from selectivity filter mutations. The relative permeability of monovalent and divalent cations against sodium for each channel tested. Values are listed in and *Figure 4—figure supplement 1*. The relative permeability ($P_x/P_{Na}$) was estimated using the Goldman-Hodgkin-Katz equation. Dashed lines indicate the lower limit of $P_x/P_{Na}$ detection under our experimental conditions (See 'Materials and methods').

The following figure supplements are available for figure 4:

**Figure supplement 1**. NsvBa can be converted into a divalent cation-selective channel.

**Figure supplement 2**. Reversal potentials ($E_{rev}$) measured at steady state with calculated relative permeability ($P_x/P_{Na}$) for bacterial Na$_v$ channels.

activation for these channels by −19 to −24 mV (−62, −55, −51 mV respectively). When tested together, temperature and basic pH effects on $V_{1/2}$ for NsvBa, NaChBac and NavBp were additive, converging on ≈−100 mV (−95, −102, −100 mV respectively). Importantly, we determined that the NsvBa remains a non-selective channel at higher pH and temperature, although the $P_x/P_{Na}$ for K$^+$ and Ca$^{2+}$ did increase slightly 2–3 times (*Figure 6—figure supplement 1*). To quantify the temperature sensitivity of these channels, the relationship between the peak current during a voltage ramp and temperature was fit to a linear equation to determine the 10-degree temperature coefficient ($Q_{10}$). The $Q_{10}$ for NaChBac and NavBp peak currents was 3.5–4.4 at neutral pH and 3.5–4.1 at basic pH. In contrast, the voltage-dependence of activation of the hNa$_v$1.1 channel was less temperature- ($Q_{10}$ = 1.2 and 1.4, *Figure 6D*) and pH- ($\Delta V_{1/2}$ < 8 mV, *Figure 5—figure supplement 1*) sensitive. Thus, pH and temperature-induced increases of the peak current and reduction of the voltage-dependence of activation are distinct from eukaryotic Na$_v$ channels.

The stimuli that depolarize these bacteria from −180 mV to the more depolarized range where voltage-gated channels activate (−40 to −100 mV, depending on pH and temperature), are not known. We speculate that $\Psi_{rest}$ declines as bacterial pumps are starved for internal H$^+$ or Na$^+$ to supply the hyperpolarizing extrusion pumps. Internal [Na$^+$] would be rapidly recharged by activation of the voltage-gated monovalent cation channels.

Since cation entry into alkaliphilic bacteria is at least partially dependent on Na$_v$ channels, we hypothesized that Na$_v$ channel antagonists would attenuate bacterial growth. Under voltage clamp, we observed that sodium current from *Bacillus* channels NavBp, NaChBac and NsvBa are blocked by known Na$_v$ channel antagonists, the local anesthetic lidocaine, the anti-hypertensive nifedipine, and the anti-estrogen tamoxifen, with similar potencies (*Figure 7A,B* and *Figure 7—figure supplement 1*). When these drugs were introduced into the culture media, growth of *Bacillus* species *alcalophilus* and *pseudofirmus* were severely impaired as measured by spectroscopic absorbance (*Figure 7C,D*). The measured half-inhibitory concentrations (IC$_{50}$) of sodium current and bacterial growth were within a half-log unit (*Figure 7—figure supplement 1*), suggesting that growth inhibition was not an off-target effect. We also examined the effect of these drugs on two matrices of *Bacillus* motility: tumbling frequency and swim speed. tamoxifen, nifedipine and lidocaine increased tumbling frequency and decreased swim speed of *B. pseudofirmus* (*Figure 7E*, *Figure 7—figure supplement 1*). However, *B. alcalophilus* swim speed was not delayed by the three drugs and only tamoxifen and nifedipine increased tumbling frequency (EC$_{50}$ = 70 µM and 375 µM, respectively).

## Discussion

We have functionally characterized NsvBa, a cation nonselective voltage-gated ion channel from *Bacillus alcalophilus*. Our findings demonstrate that this family of prokaryotic voltage-gated Na$^+$ channels are not exclusively Na$^+$-selective and that the filter sequence that controls passage of cations into the bacteria are transferable within the *Bacillus* genus. Through a series of mutations, we showed that the selectivity of *Bacillus* Na$_v$ channels can be converted from nonselective, to relatively higher Na$^+$ or Ca$^{2+}$ selectivity, suggesting that the filter is readily mutable in evolution to adapt to ionic conditions. These highly adaptable selectivity filters are critical for the *Bacillus* alkaliphiles, allowing for the

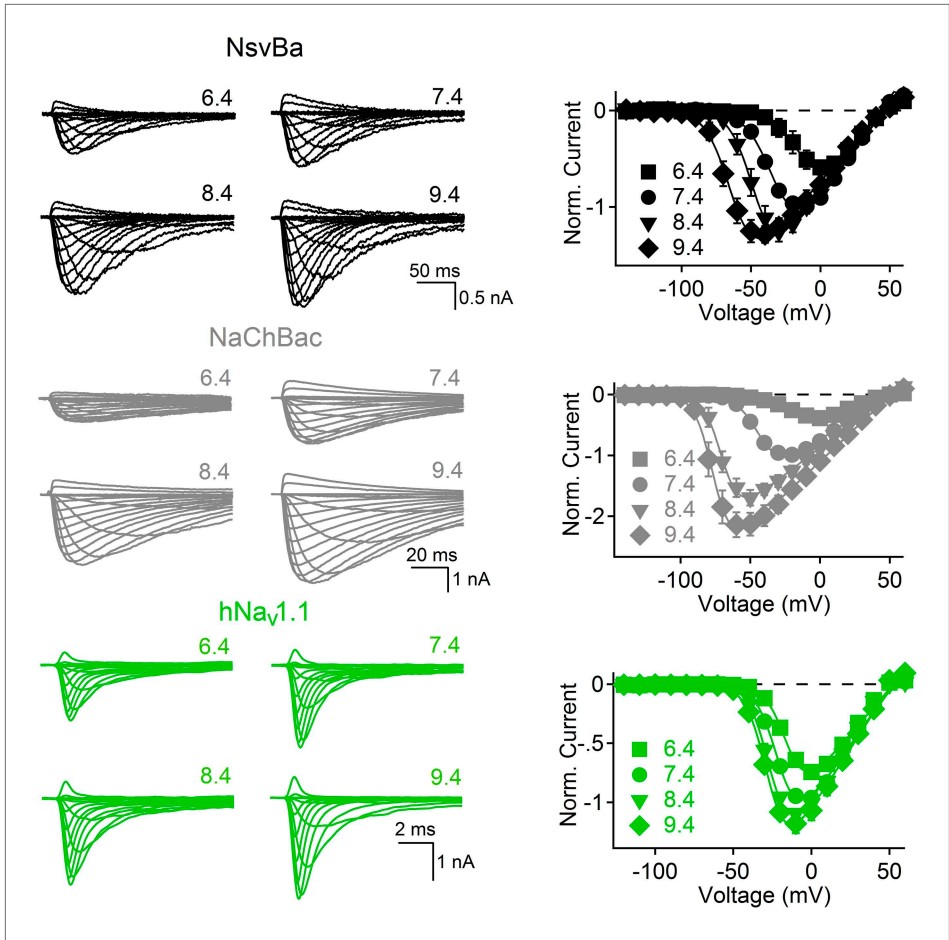

**Figure 5**. The bacterial Na$_v$ channels are modulated by extracellular alkaline pH (pHo). *Left*, representative traces recorded from one cell expressing the NsvBa (black), NaChBac (gray) and hNav1.1 (green) channels in 150 mM NaCl conditions with the external pH adjusted to 6.4, 7.4, 8.4 and 9.4. Currents were activated by +10 mV steps from a holding potential of −140 mV (NsvBa and NaChBac) or −120 mV (hNav1.1). Right, resulting voltage current relationship normalized to the peak current measured in the 7.4 pHo condition (n = 4–7 for each channel, Error = ±SEM).

The following figure supplement is available for figure 5:

**Figure supplement 1**. Na$_v$ steady state voltage-dependence of activation (V$_{1/2}$) measured in different extracellular pH (pH$_o$) conditions.

habitation of various cation-rich environments in which they presumably evolved. It is anticipated that the ion-specificity of at least some antiporters will parallel that of the major coupling ions for the voltage-gated channels that provide a significant amount of their cytoplasmic substrate.

Although no Na$^+$ ions reside within the filter of the wild type NavAb crystal structures, 3 Na$^+$ coordinating sites were proposed: glutamate side chains form a high-field-strength site (Site$_{HFS}$) near the extracellular end of the filter, while backbone carbonyls of Leu and Thr comprise central (Site$_{CEN}$) and inner sites (Site$_{IN}$) (*Payandeh et al., 2011*). In the CavAb structure, 3 hydrated Ca$^{2+}$ ions were found coordinated within the filter and 3 sites (Sites 1–3) proposed within the filter. Site 1, the one closest to the extracellular surface, is formed by 4 Asp carboxyl side chains, equivalent to positions Ser 196 in NaChBac and Ser 190 in NsvBa. Site 2 (equivalent to Site$_{HFS}$ in the wt NavAb crystal structure) is formed by four side chain carboxyls and four backbone carbonyls from equivalent residues of Glu 192 in NaChBac and Asp 186 in NsvBa. Our data suggest that the shorter side chain of the NsvBa Asp acidic residue at site 2 is correlated with decreased Na$^+$-selectivity, but Na$^+$ selectivity can be artificially endowed when replaced by a Glu (D186E) (*Figure 3—figure supplement 1*). In agreement with this interpretation, a recent report demonstrates that when the longer side chain Glu 192 is mutated to the

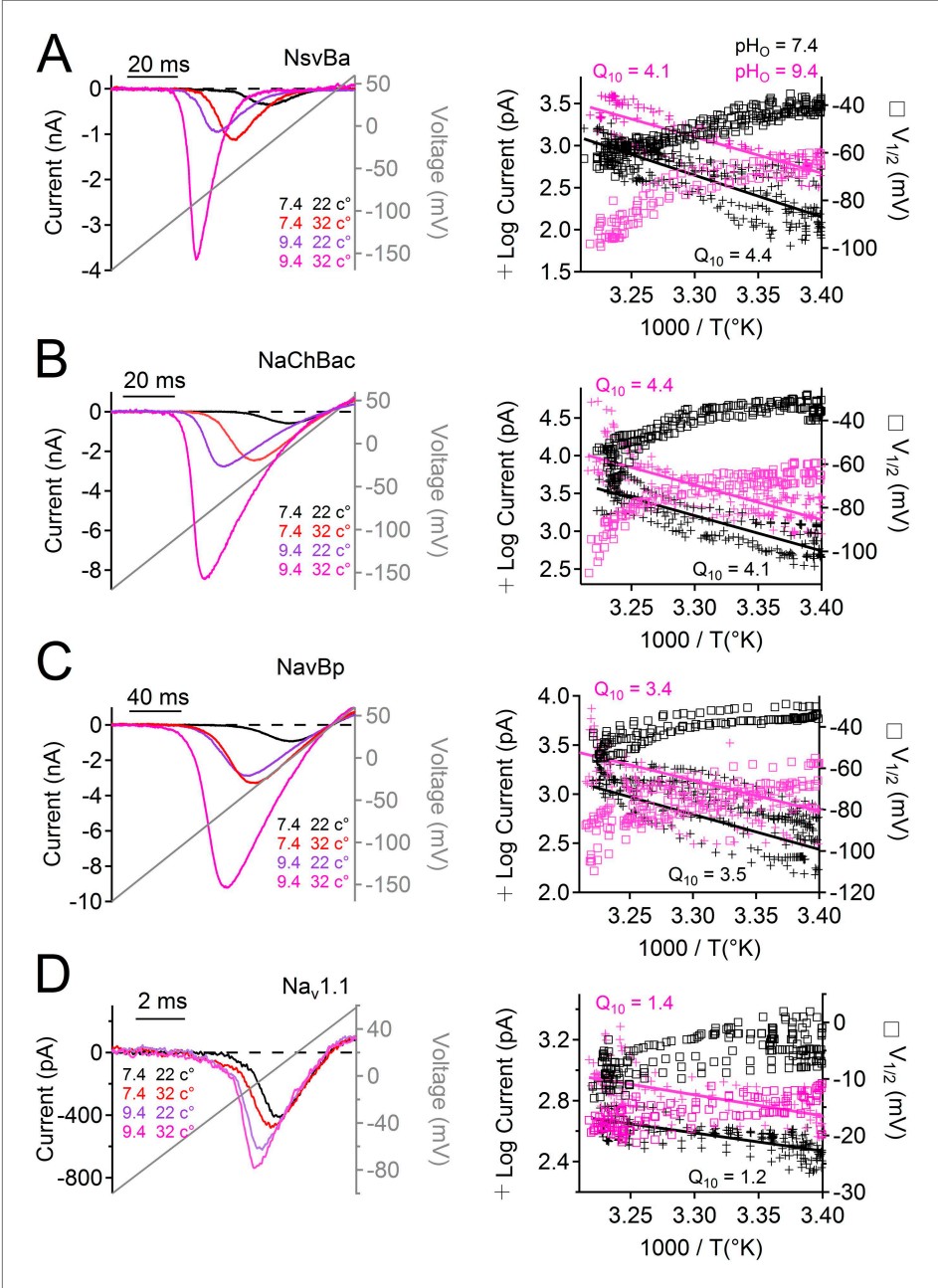

**Figure 6**. Temperature and pH dependence of sodium channels. *Left* Example I$_{Na}$ traces conducted by (**A**) NsvBa, (**B**) NaChBac, (**C**) NavBp and (**D**) Na$_v$1.1 channels, when the extracellular saline (pH = 7.4 or 9.4) was heated from 20 to 37°C. Channels were activated by a 0.5 Hz voltage ramp. Voltage ramps were applied for different durations to compensate for different channel kinetics of activation and inactivation: NsvBa and NaChBac (100 ms); NavBp (200 ms) and Na$_v$1.1 (10 ms). *Right*, Arrhenius plots with resulting peak current (plus symbols) and V$_{1/2}$ (open squares) are graphed as a function of temperature. The peak currents were fit to a linear equation and the resulting slope (Peak Q$_{10}$) given for both external pH conditions (n = 4).

The following figure supplement is available for figure 6:

**Figure supplement 1**. The effect of temperature and pH on NsvBa selectivity.

shorter Asp in NaChBac (E192D), the channel becomes less selective for Na$^+$ among Ca$^{2+}$ and K$^+$ ions (**Finol-Urdaneta et al., 2014**). The full-length bacteria Na$_v$ from marine α-proteobacterium *Rickettsiales* HIMB114 (NavRh) was crystallized containing the selectivity filter TLSSWET-. Although the NavRh

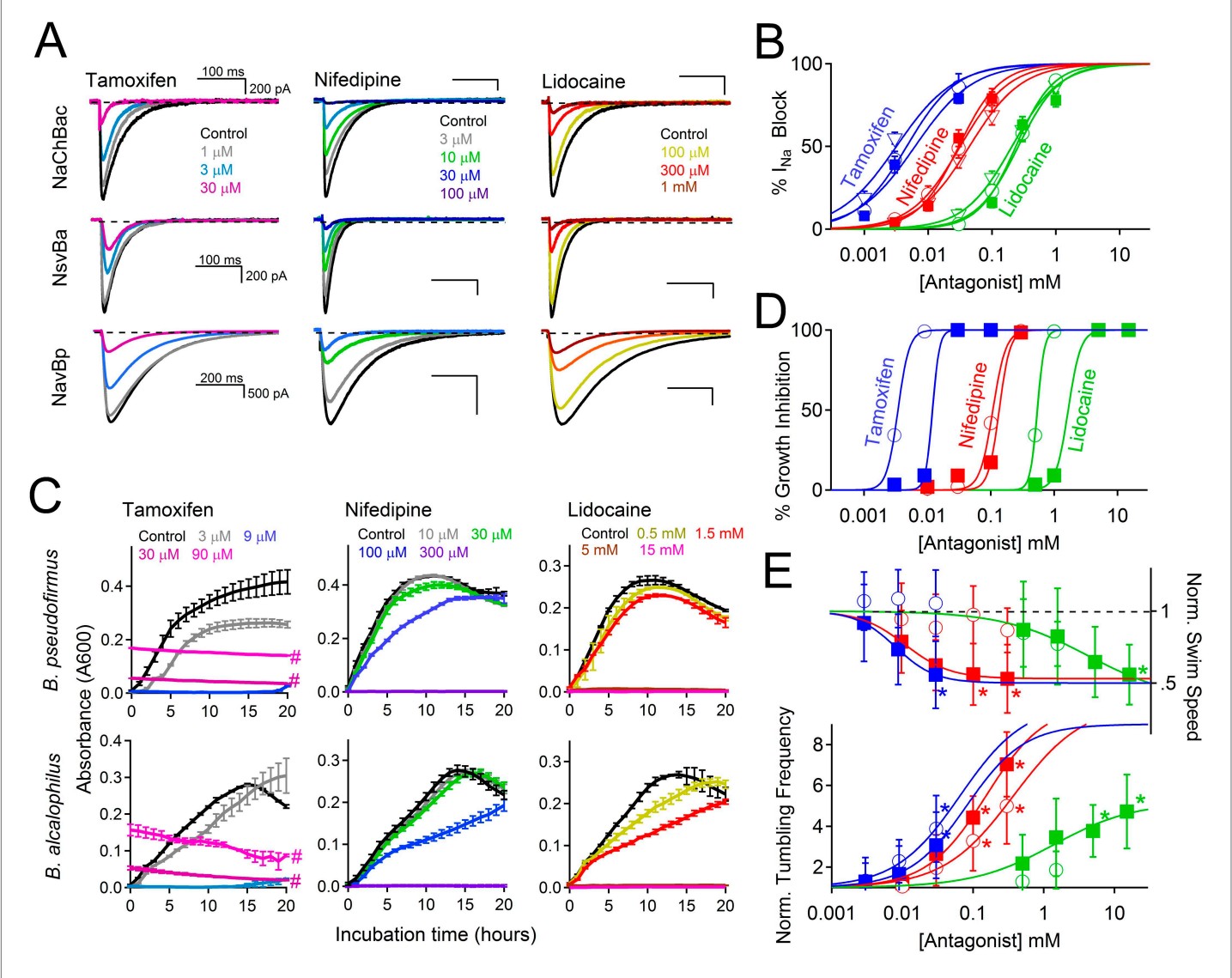

**Figure 7**. Antagonism of Na$_v$ channels blocks the growth and motility of alkaliphilic Bacillus. (**A**) Example Na$^+$ currents from NaChBac, NsvBa and NavBp channels in the presence of vehicle control (≤0.1% DMSO) and extracellularly applied drugs. Currents were activated by a 0.2 Hz train of 500 ms depolarizations to 0 mV from −140 mV. (**B**) The resulting NsvBa (open circles) and NaChBac (filled triangles) Na$^+$ current block-drug relationship by tamoxifen, nifedipine and lidocaine (n = 4–6 cells per concentration, Error = ±SEM). (**C**) The effect of drugs on the time course of bacterial growth. (**D**) The resulting drug antagonism of growth by *Bacillus alcalophilus* (open circles) and *Bacillus pseudofirmus* (filled squares) inferred by light spectroscopy 600 λ absorbance (n = 4 growth trials, Error = ±SEM). Tamoxifen is a fluorescent compound which significantly absorbs 600 λ light at concentrations above 30 μM, indicated by a hashtag (#). (**E**) The effect of nifedipine, tamoxifen, and lidocaine on *Bacillus alcalophilus* (open circles) and *pseudofirmus* (filled squares) tumbling and swim speed (n = 4 trials, Error = ±SEM). Statistical significance of motility in untreated and drug-treated cells using a Student's t-test and indicated by an asterisk (p < 0.05) (*). EC$_{50}$ and IC$_{50}$ values for all assays are listed in *Figure 7—figure supplement 1*.

The following figure supplement is available for figure 7:

**Figure supplement 1**. The effect of Na$_v$ antagonist on I$_{Na}$, bacteria motility and growth by drugs: lidocaine, tamoxifen and nifedipine.

channel did not function when heterologously expressed in mammalian and insect cells, the filter was found to be Na$^+$-selective when grafted into the NaChBac channel. It is surprising that although the glutamates within the NavRh and NavAb filters originate at distinct positions (See *Figure 1*), the carboxyl side chains from both channels occupy the same positions in space (*Zhang et al., 2012*). Thus, it is likely that selectivity for Na$^+$ over other cations is partly achieved by the geometry of the

arrangement of the carboxylates at Site 2 (Site$_{HFS}$) within the filter of bacterial Na$^+$ channels. High-resolution crystal structures of Na$_v$ channels in which sodium ions are resolved are needed to confirm this hypothesis.

Among all sequenced living species, naturally occurring nonselective voltage-gated ion channels are rare. It is not clear whether cation-selective or nonselective filters arose first within prokaryotes. Analysis comparing domains of bacteria and mammalian Na$_v$ channels indicate that Na$^+$-selectivity was independently acquired in these channels families (*Liebeskind et al., 2013*). Bacterial Na$_v$ channels are functionally more related to homotetrameric CatSper and Kv channels and thus may not be direct ancestors of mammalian Na$_v$ channels. Our results are consistent with this interpretation. Not surprisingly, we were not able convert *Bacillus* Na$_v$ channels to K$^+$-selective channels by mutating the filter sequence, presumably since the pore architecture between Kv and bacterial Na$_v$ channels (e.g. lumen size, filter length and number of pore helices) are structurally divergent. Our results suggest that the 'GxG motif' found in the NsvBa filter evolved convergently within the bacterial Na$_v$ family and is only distantly related to K$^+$-selective Kv channels. A recent report on the expression and characterization of Na$^+$ channel homologs from the invertebrate sea anemone *Nematostella vectensis* (NvNa$_v$2.1), revealed a heterotetrameric Na$^+$ channels bearing noncanonical selectivity site (DEEA) that was not selective among K$^+$ and Na$^+$ ions (*Gur Barzilai et al., 2012*). Thus it appears that the homomeric prokaryotic NsvBa and functionally-heteromeric eukaryotic NvNa$_v$2.1 channels are examples of evolution of Na$^+$-selectivity to non-selectivity that occurred independently within prokaryotes and metazoans.

In excitable eukaryotic tissues such as nerves and muscle, the half-activation threshold (V$_{1/2}$) of voltage-gated sodium channels is within ~80 mV of their cellular resting membrane potential (Ψ$_{rest}$ ~ -60 to −90 mV). At first glance, this appears to contrast with the 140 mV difference between the Ψ$_{rest}$ and V$_{1/2}$ of alkaliphilic *Bacillus*. But as we have shown, increasing alkalinity and temperature shift bacterial Na$_v$ V$_{1/2}$'s to approximately −100 mV. At temperatures approaching 40°C, our patch recordings became unstable, thus limiting the temperature range we tested to <37°C. If the sodium current sensitivity is extrapolated to 60°C, the voltage dependence would likely be well within 80 mV of the most negative bacterial resting membrane potential. Since bacterial Ψ$_{rest}$ is a function of metabolic state (Na$^+$/H$^+$ pumping) and bacteria occupy more variable ionic environments than neurons, the larger difference between 'optimal' a Ψ$_{rest}$ and V$_{1/2}$ makes physiological sense. The high temperature and pH sensitivity of bacterial Na$_v$ channels is similar to TRP and some Kv channels (*Patapoutian et al., 2003*; *Yang and Zheng, 2014*), and is not shared by vertebrate Na$_v$ channels. We suspect that the fidelity of neuronal firing in metazoans, which depend on their Na$_v$ channels, led to evolution of less sensitivity to pH and temperature, perhaps by selection of the residues that become exposed to waters within the voltage-sensing S4 and selectivity filter (*Chowdhury et al., 2014*). In contrast, we propose that alkaliphilic bacteria, which dwell in high temperature (35–60°C) and high pH (9–11) environments, employ homomeric Na$_v$ channels for an entirely different purpose, that of rapidly adjusting internal sodium concentration for metabolic control. This would thus impact growth rates of these *Bacillus* species as we have observed.

Concentrations of lidocaine, nifedipine, and tamoxifen that inhibit heterologously-expressed bacterial Na$_v$ channels (NavBp and NsvBa) also block the growth of native *Bacillus species* (*B. pseudofirmus* and *B. alcalophilus*). Although all three drugs affected motility for *B. pseudofirmus*, only tamoxifen and nifedipine increased tumbling frequency of *B. alcalophilus*. These results suggest that in *B. alcalophilus* there may be other transporters or channels besides NsvBa that provide a cation source to drive flagellar motion. These finding demonstrate that block of Na$^+$ entry via Na$_v$, which disrupts the sodium cycle, can impair *Bacillus* motility and growth. The prokaryotic Na$_v$ from *Magnetococcus marinus* (NavMs), was recently crystalized with a brominated drugs bound and the proposed binding site validated (*Bagneris et al., 2014*). Thus voltage-gated channels found in pathogenic *Bacillus* (e.g. *Bacillus cereus* and *anthracis*) could potentially be antimicrobial targets.

# Materials and methods

## Whole-cell voltage-clamp experiments

HEK 293T cells were transiently transfected with mammalian cell expression plasmid pTracer CMV2 containing either NsvBa or NaChBac genes. The NCBI GenBank accession number for NsvBa is JX399518.1 and is annotated as a K$^+$ ion transporter (BalcAV3624). Cells were seeded onto glass coverslips, and placed in a perfusion chamber for experiments where extracellular conditions could be

altered. With the exception of the experiments described in *Figure 5*, *Figure 2—figure supplement 1* and *Figure 5—figure supplement 1*, the pipette electrode solution contained (in mM): NMDG (90), NaCl (20), HEPES (10), EGTA (5), CaCl$_2$ (0.5) and pH was adjusted to 7.4 with HCl. When testing the relative permeability of monovalent cations, the bath solution contained (in mM): X-Cl (150), HEPES (10), EGTA (5), CaCl$_2$ (0.5) and the pH was adjusted with X-OH, where the X is the indicated monovalent cation. When testing the relative permeability of divalent cations, the bath solution contained: X-Cl$_2$ (110), HEPES (10), EGTA (5), CaCl$_2$ (0.5) and the pH was adjusted with X-(OH)$_2$, where the X is the indicated divalent cation. For the experiments in *Figure 5*, extracellular saline contained NaCl (150), HEPES (10), CaCl$_2$ (2) and pH was adjusted with CsOH so that the concentration of permeant Na$^+$ ion remained constant at high pH. All saline solutions were adjusted to 300 mOsm (±5) with mannitol, if needed. Data were analyzed by Igor Pro 7.00 (Wavemetrics, Lake Oswego, OR). Residual leak (>−100 pA) and capacitance were subtracted using a standard P/4 protocol. Current-voltage-relationships were fit with the following equation:

$$I = \frac{V - E_{rev}}{\{1 + \exp\left(\dfrac{V - V\,\frac{1}{2}}{RT/F}\right)}$$

where I is current and $E_{rev}$ is the extrapolated reversal potential. $E_{rev}$ was used to determine the relative permeability of monovalent cation X to Na ($P_x/P_{Na}$) according to the following equation (*Hille, 1972*; *Sun et al., 1997*).

$$\frac{P_x}{P_{Na}} = \frac{\alpha_{Nae}}{\alpha_{xe}}\left[\exp\left(\frac{\Delta E_{rev}}{RT/F}\right)\right]$$

where $\Delta E_{rev}$, $\alpha_x$, R, T and F are the reversal potential, effective activity coefficients for cation x (i, internal and e, external), the universal gas constant, absolute temperature, and the Faraday constant, respectively. For these pseudo-bionic conditions, we are assuming that the internal NMDG is impermeant—it has a similar ionic radius as cesium, which is impermeant in all of the Nav channels we tested. The effective activity coefficients ($\alpha_x$) were calculated using the following equations:

$$\alpha_x = \gamma_x [X]$$

where $\gamma_x$ is the activity coefficient and [X] is the concentration of the ion. For calculations of membrane permeability, activity coefficients ($\gamma$) was calculated using the Debye-Hückel equation: 0.74, 0.76, 0.72, 0.71, 0.69, 0.35, 0.29, 0.27 and 0.27 correspond to Na$^+$, Li$^+$, K$^+$, Rb$^+$, Cs$^+$, Mg$^{2+}$, Ca$^{2+}$, Sr$^{2+}$ and Ba$^{2+}$ respectively. To determine the relative permeably of divalent cations to Na$^+$, the following equation was used:

$$\frac{P_x}{P_{Na}} = \frac{\{\alpha_{Nai}\left[\exp\left(\dfrac{E_{rev}F}{RT}\right)\right][\exp\left(\dfrac{E_{rev}F}{RT}\right) + 1]\}}{4\alpha_{xe}}$$

$E_{rev}$ for each cation condition was corrected to the measured liquid junction potentials (−4.4 to 3.4 mV). In some cationic conditions, no inward (negative) voltage-dependent currents could be activated, but $E_{rev}$ was measured as ≤−4 mV. In these cases, the lower limit of $P_x/P_{Na}$ was reached (0.1) due to low levels of endogenous chloride and nonselective currents in HEK cells effecting $E_{rev}$. The voltage-dependence of channel activation and inactivation was fit to the equation:

$$I_{norm} = I_{max} - (I_{norm} - I_{min})/(1 + \exp\left[\frac{V - V\,\frac{1}{2}}{k}\right])$$

where $I_{max}$ and $I_{min}$ are the maximum and minimum current values, V is the applied voltage, $V_{1/2}$ is the voltage at half activation, and k is the slope factor. For the single channel conductance measurements, experimental conditions were the same except that the pipette and bath saline solutions were switched. The equation for the exponential fits used in to estimate the rate of inactivation was:

$$f(x) = B + A\,exp[\left(\frac{1}{\tau}\right)x]$$

where τ is the time constant.

For the experiments performed in *Figure 6*, where extracellular pH and/ or temperature were altered, the pipette electrode solution contained (in mM): CsMES (100), HEPES (10), NaCl (15), EGTA (6), TAPS (5), $MgCl_2$ (2), $CaCl_2$ (3) and pH was adjusted to 6.4–9.4 with CsOH or HCl. For temperature-controlled experiments, the perfusate was heated and cooled at rate of 0.4–2°C / s using a Warner TC-344B heater controller and Warner SHM-6 solution heater while bath temperature was monitored using a thermistor placed in close proximity to the recording electrode. A linear fit of the peak currents during the voltage ramp was used to determine $Q_{10}$, as described by the Arrhenius equation:

$$Q_{10} = \left[\frac{R2}{R1}\right]^{10/(T2-T1)}$$

Where R = rate and T = temperature

## Alkaliphilic bacterial growth assays

*B. alcalophilus* and *B. pseudofirmus* OF4 were grown in malate yeast extract (MYE) or KMYE (50 mM $K_2CO_3$ instead of $Na_2CO_3$ in MYE) medium with shaking at 37°C for 6 hr. Cells were suspended in 1.0 ml of the MYE or KMYE medium (with or without 10, 30, 100, or 300 µM nifedipine), and incubated at 37°C for 10 min. Microscopic observation was carried out immediately by the hanging drop method using a Leica DMLB100 dark field microscope (400×) and Leica DC300F camera, Leica IM50 version 1.20 software (Leica Geosystems, Tokyo), and recorded with Display capture ARE software (http://www.vector.co.jp/soft/win95/art/se221399.html). The swimming speed of 40 individual cells (swimming for more than 15 s), and swimming fractions of more than 50 individual cells were measured by 2D movement measurement capture 2D-PTV software (DigimoCo., Ltd.). All results shown are the averages of three independent experiments. Drug antagonism of bacteria growth was assessed using the equation:

$$\frac{Ab_{control} - Ab_{drug}}{Ab_{control}} \times 100$$

where $Ab_{control}$ and $Ab_{drug}$ are the maximum spectroscopic absorbance at 600λ measured within 16 hr of bacterial growth in untreated and drug treated conditions, respectively. The growth inhibition-drug concentration relations were fit using the Hill equation described above.

## Acknowledgements

The authors would like to thank their respective funding agencies and grants: PGD was supported by NIH T32 HL007572; R01 GM028454 to TAK; A Strategic Research Foundation Grant-in-Aid for Private Universities and a Grant-in-Aid for Scientific Research on Innovative Areas No. 24117005 of the Ministry of Education, Culture, Sports, Science and Technology of Japan were awarded to M.I

## Additional information

### Funding

| Funder | Grant reference number | Author |
| --- | --- | --- |
| National Institutes of Health | R01 GM028454 | Terry A Krulwich |
| National Institutes of Health | T32 HL007572 | Paul G DeCaen |
| Ministry of Education, Culture, Sports, Science, and Technology | 24117005 | Masahiro Ito |

The funders had no role in study design, data collection and interpretation, or the decision to submit the work for publication.

## Author contributions

PGD, Conception and design, Acquisition of data, Analysis and interpretation of data, Drafting or revising the article; YT, Acquisition of data, Contributed unpublished essential data or reagents; TAK, Drafting or revising the article, Contributed unpublished essential data or reagents; MI, Conception and design, Analysis and interpretation of data; DEC, Conception and design, Drafting or revising the article

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
