## [Decision Letter]

Thank you for sending your work entitled “Ionic selectivity and thermal adaptations within the voltage-gated sodium channel family of alkaliphilic Bacillus” for consideration at *eLife*. Your article has been favorably evaluated by Vivek Malhotra (Senior editor) and 3 reviewers, one of whom is a member of our Board of Reviewing Editors.

The following individuals responsible for the peer review of your submission have agreed to reveal their identity: Richard Aldrich (Reviewing editor); Youxing Jiang and Baron Chanda (peer reviewers).

The Reviewing editor and the other reviewers discussed their comments before we reached this decision, and the Reviewing editor has assembled the following comments to help you prepare a revised submission.

In recent years, prokaryotic sodium channels have become model systems to study the structural basis of sodium ion transport and selectivity. In this study, DeCaen et al. describe a new member of this family that lacks the selectivity of the canonical prokaryotic sodium channels. They determined the permeation and gating properties of NsvBa and demonstrated that the channel is nonselective among Na^+^, Ca^2+^ and K^+^. Based on the filter sequence difference between NsvBa and other Na^+^ selective prokaryotic Nav channels, the authors performed mutagenesis and demonstrate that the nonselective filter from NsvBa can be converted into one that is selective for Na^+^ or Ca^2+^, and that these features can be conferred onto the NaChBac filter. The authors also characterized the voltage-, pH- and temperature-dependence of several Nav channels from several Bacillus species and demonstrated that a combination of higher temperature and pH are required to reduce the activation threshold of channel opening in these alkaliphilic Bacillus bacteria. This lends credence to their claim that these channels are involved in transport of sodium and other alkali metal ions at highly hyperpolarized potentials which is also the resting membrane potential of this bacterial host. In addition, the authors also characterized the pharmacological properties of Bacillus Nav channels using various Nav channel antagonists and demonstrated that these drugs inhibit the cell growth likely due to the block of Na influx through Nav channels.

Like this group's previous studies on several prokaryotic Na channels, including the cloning and characterization of the 1st prokaryotic Nav channel (NaChBac) also from a Bacillus species, the characterization of NsvBa presented here is systematic and provide important insights into the selectivity and gating of the Nav channel family. In addition, this group's discovery and characterization of several members of prokaryotic homo-tetrameric Nav channel family have provided excellent model systems for structural studies of Na/Ca channels. This work is well done and nicely presented.

The only substantive concern is regarding the claim that these channels are non-selective for sodium. All the selectivity data shown here is based on experiments at neutral pH and 22 degree Celsius. We would like to see if this non-selectivity is maintained at higher temperatures and at alkaline pHs. If these channels remain truly non-selective under those conditions, then one wonders what is the role of sodium selective voltage-gated channels?

---

## [Author Response]

*[…] The only substantive concern is regarding the claim that these channels are non-selective for sodium. All the selectivity data shown here is based on experiments at neutral pH and 22 degree Celsius. We would like to see if this non-selectivity is maintained at higher temperatures and at alkaline pHs. If these channels remain truly non-selective under those conditions*, *then one wonders what is the role of sodium selective voltage-gated channels?*

We tested P_x_/P_Na_ of K^+^ and Ca^2+^ at pH_o_ = 9 and at 34^°^C for the NsvBa channel at steady state and added this data to the manuscript (Figure 6—figure supplement 1). These are exceptionally difficult conditions, as whole cell patches become very unstable when held at -140 mV for five to ten minutes. The differences in E_rev_’s measured at neutral pH/22⁰C and basic pH/34⁰C are not different for Na^+^ conditions but do increase slightly (≤ 5 mV) in the K^+^ and Ca^2+^ conditions. This translates to a 2-3 fold increase the relative permeability for K^+^ and Ca^2+^, which is interesting but does not achieve the level of selectivity found in Ca_v_s or Kv channels. Thus, we are confident in our conclusion that NsvBa is a non-selective channel, as there will always be slight differences of ion preference within this selectivity classification.

Additional data added to the manuscript:

We added data to Figure 7 (lidocaine and tamoxifen effects in bacteria motility assays), which was not available at the time of submission. These drugs inhibit swim speed and increase tumble frequency for B. pseudofirmus but the results are mixed for B. alcalophilus. Our interpretation is that B. alcalophilus may have other transporters or channels besides NsvBa that provide a cation source to drive flagellar motion. We have discussed the result in the text.

Additional edits to the manuscript:

1) Included references (Yang et al ; Chaudhury et al.) to Kv channel temperature sensitivities that were published during review.

2) Figure 5—figure supplement 1 is now Figure 5, bringing the total of main text figures to 7.

3) The NCBI Genbank number for NsvBa was added to the Methods.